# Addressing The False Negative Problem of Deep Learning MRI Reconstruction Models by Adversarial Attacks and Robust Training

**Kaiyang Cheng** * [1,2]                                     VICTORCHENG21@BERKELEY.EDU

**Francesco Calivá** *[2]                                     FRANCESCO.CALIVA@UCSF.EDU

**Rutwik Shah**[2]                                           RUTWIK.SHAH@UCSF.EDU

**Misung Han**[2]                                            MISUNG.HAN@UCSF.EDU

**Sharmila Majumdar**[2]                                     SHARMILA.MAJUMDAR@UCSF.EDU

**Valentina Pedoia**[2]                                      VALENTINA.PEDOIA@UCSF.EDU

[1] *Department of Electrical Engineering and Computer Sciences, University of California, Berkeley*
[2] *$CI^2$, Center for Intelligent Imaging, Department of Radiology and Biomedical Imaging, University of California, San Francisco*

## Abstract

Deep learning models have been shown to be successful in accelerating MRI reconstruction, over traditional methods. However, it has been observed that these methods tend to miss rare small features, such as meniscal tears, subchondral osteophyte, etc. in musculoskeletal applications. This is a concerning finding as these small and rare features are the particularly relevant in clinical diagnostic settings. Additionally, such potentially dangerous loss of details in the reconstructed images are not reflected by global image fidelity metrics such as mean-square error (MSE) and structural similarity metric (SSIM). In this work, we propose a framework to find the worst-case false negatives by adversarially attacking the trained models and improve the models'ability to reconstruct the small features by robust training.

**Keywords:** MRI Reconstruction, Adversarial Attack, Robust Training.

## 1. Introduction

High data quality is a priority in medical image analysis. Magnetic resonance imaging (MRI) has the capability of satisfying such requirement when it comes to screening soft tissues. Nevertheless, MRI has a limitation of requiring long scanning time. As a consequence, over the past few years, acceleration of MRI has received an increasing level of attention, which has not been restricted only to medical physicist but extended also to the deep learning community. The book chapter from Hammernik and Knoll (2020), the Fast-MRI challenge that was held at NeurIPS 2019 (Zbontar et al., 2018), the AccelMR 2020 and MC-MRRec challenges at ISBI and MIDL 2020, in addition to dedicated sessions at international con-

---

* Contributed equally

ferences such as ISMRM, MICCAI are all examples that deep learning-powered accelerated MRI is an active topic of research.

### 1.1. Hypotheses for false negative

Despite a remarkable improvement in image quality of accelerated MRI from Deep Learning-based methods, the false negative reconstruction phenomenon is still present. The false negative phenomenon of MRI reconstruction refers to qualitative observations provided during the announcement of NeurIPS 2019 FastMRI challenge results[1]. The top performing models in terms of structural similarity metric (SSIM) and radiologists' image quality assessment were shown to have failed in reconstructing some relatively small abnormalities such as meniscal tear and subchondral osteophyte. In the attempt to better explain this phenomenon we investigate two hypotheses:

1) the information of small abnormality features is completely lost through the under-sampling process;

2) the information of small abnormality features is not completely lost. Instead, it is attenuated and laid in the tail-end of the distribution, hence is rare.

Were the first hypothesis true, it would be impossible for any method to reconstruct a small abnormality feature, unless the presence of the abnormality is confounded with other structural changes. We are unable to formally verify whether this hypothesis is always false. Nonetheless, we are able to demonstrate that the condition stated in hypothesis 1 is unlikely to occur. Were the second hypothesis true, it would be possible for a reconstruction model to reconstruct it. This is especially true with data-driven and learning-based methods. In this work, we show that the second hypothesis is true in many cases, and it is possible for a deep learning reconstruction model to reconstruct the small abnormality by using the limited information available.

To investigate these two hypotheses, we define '*false-negative adversarial feature*' (FNAF), a perceptible small feature which is present in the ground truth MRI but has disappeared upon MRI reconstruction, which is performed via a learning model. The contributions of this paper can be summarized as follows:

1) We quantitatively show that highly performing deep learning reconstruction models trained to only maximize image quality can fail when it comes to reconstructing small and infrequent structures.

2) We quantitatively show that it is possible to reconstruct small structures if the right set of priors is available during training, particularly if adversarial training is employed.

## 2. Related works

### 2.1. MRI Reconstruction with Deep Learning

MRI reconstruction from undersampled k-space is key in fast MRI (Liang et al., 2019; Hammernik and Knoll, 2020). Liang et al. (2019) explains that deep learning-powered

---

1. https://slideslive.com/38922093/medical-imaging-meets-neurips-4

MRI reconstruction can be accomplished following either data-driven, model-driven or integrated approaches. Data-driven approaches are generally data hungry and do not require prior knowledge, mainly because they take advantage of a huge amount of data to learn the mapping between raw data and the reconstructed MRI. In model-based approaches, the solution space is restricted by injecting task prior knowledge. This can be obtained for instance by reproducing the iterative approach of compressed sensing. Integrated approaches combine positive aspects of both previous solutions.

## 2.2. Adversarial attack by small perturbation

To apply the practice of adversarial attack to MRI reconstruction with deep learning, it is important to understand the most studied forms of adversarial attack, adding small imperceptible perturbations to input images with the aim to mislead machine learning models (Biggio et al., 2013; Szegedy et al., 2014; Goodfellow et al., 2015). Goodfellow et al. (2015); Bubeck et al. (2018); Gilmer et al. (2018); Mahloujifar et al. (2019); Shafahi et al. (2019a) attempts to develop a variety of theories that could explain these adversarial examples. One notable theory is that adversarial examples are a consequence of data scarcity (Schmidt et al., 2018), as the true data distribution is not being captured by non-sufficiently large dataset. Another profound explanation is provided by Ilyas et al. (2019), which shows that adversarial successes are mainly supported by model's ability to generalize on standard test set by using non-robust features. In other words, adversarial examples are more likely a product of datasets rather than that of machine learning models. To make a model resistant to adversarial attacks without additional data, one could employ adversarial training and provide the model with a prior that remarks the fact that non-robust features are not useful (Goodfellow et al., 2015; Madry et al., 2018). These findings are orthogonal to the second investigated hypothesis: if we interpret the distribution of FNAF as the distribution of robust features, we may attribute FNAF reconstruction failure to the dataset's inability to capture FNAF's distribution.

## 2.3. Adversarial attack on generative networks

While most of adversarial attacks focus on discriminative models, Kos et al. (2017) propose a framework to attack variational autoencoders (VAE) and the VAE-GAN. Specifically, input images are imperceptibly perturbed so that the generative models generate target images that belong to a different class. Although reconstruction models can be seen as generative, we differ from this work, mainly because we focus on generating perceptible features that perform un-targeted attacks.

## 2.4. Adversarial attacks via bi- and three- dimensional transformations or physical attacks

Going beyond small perturbations, a set of more realistic attacks produced by 2D and 3D transformations has been proposed in Xiao et al. (2018); Athalye et al. (2018). Similarly to our work, these studies perform perceptible attacks. Arguably, the most realistic attacks are physical attacks, which are achieved by altering the physical space before an image is captured digitally (Kurakin et al., 2017). Kügler et al. (2018) propose a physical attack on Dermoscopy images by drawing on the skin, around areas of interest. Although these

attacks could more easily translate to real world scenarios, it would be nearly impossible to perform physical attacks with imaging modalities such as MRI.

Previously described adversarial attacks utilize the fact than certain small perturbations, spatial and textual transformation in the digital and/or physical world, do not alter the image semantics. This work utilize the fact that MRI reconstruction models should reconstruct all features of an under-sampled image.

## 3. Methods

### 3.1. False negative adversarial attack on reconstruction networks

Adversarial attacks aim to maximize the loss $\mathscr{L}$ of a machine learning model, parameterized by $\theta$. This can be achieved by changing a perturbation parameter $\delta$ within the set $S \subseteq R^d$ of the allowed perturbation distribution (Madry et al., 2018) – which we restrict to be a set of visible small features in all the locations of an image. This can formally be expressed as:

$$\max_{\delta \in S} \mathscr{L}(\theta, x + \delta, y) \tag{1}$$

$\mathscr{L}$ can be any arbitrary loss function. To apply Equation (1) to reconstruction, the reconstruction network aims to reconstruct all the features including the perturbation (small features). Conversely, the attacker aims to find the perturbation (small features) which the network is not capable of reconstructing.

Let $\delta$ be an under-sampled perturbation which is added to an undersampled image and $\delta'$ the respective fully-sampled perturbation, the objective function becomes:

$$\max_{\delta \in S} \mathscr{L}(\theta, x + \delta, y + \delta') \tag{2}$$

with:

$$\delta = U(\delta') \tag{3}$$

$U$ can be any under-sampling function, comprised of an indicator function $M$, which acts as a mask in the k-space domain, and an operator that allows for a conversion from image to k-space and vice-versa such as the Fast Fourier Transform (FFT) $\mathcal{F}$ and the inverse-FFT $\mathcal{F}^{-1}$. The under-sampling and the k-space mask $M$ functions are the same as the implementations provided by Zbontar et al. (2018).

$$U(y) = \mathcal{F}^{-1}(M(\mathcal{F}(y))) \tag{4}$$

Since we synthetically construct the small added features, we can measure the loss value within the area occupied by each features and be aware if the features are reconstructed. In practice, we place a mask on the reconstructed image and the perturbed target image, so that only the area of the small feature is highlighted. The area is relaxed so that a small region at a distance $d$ from the feature border is also included. The motivation for the mask accounting for boundaries is: if only the loss of the FNAF's foreground is measured, this might not capture some failure cases where the FNAF had blended-in with the background.

Therefore, the loss is computed in a 5 pixels distance range from the boundary of the FNAF. The loss is defined as

$$\mathcal{L} = \alpha \cdot MSE(x, y) + \beta \cdot MSE(T(x), T(y)) \tag{5}$$

where x and y are the original and reconstructed MRIs respectively. T is an indicator function which masks over the FNAF in the ground truth and the reconstructed images. Weights $\alpha$ and $\beta$ are hyper-parameters set to 1 and 100 during adversarial training (described in details in section 4.2) . This allows one to better preserve both image quality and robustness of FNAF. Conversely, during attack evaluation, $\alpha$ and $\beta$ are set to 0 and 1, to only evaluate FNAF reconstruction. The loss is maximized by either random search or finite-difference approximation gradient ascent.

### 3.1.1. RANDOM SEARCH

We generate random shapes of feature $\delta$ at random locations in the image and find the $\delta$ that maximizes the loss in Equation (2). Random search (Bergstra and Bengio, 2012) has been shown to be an effective optimization technique.

### 3.1.2. FINITE-DIFFERENCE APPROXIMATED GRADIENT ASCENT

We notice that the location of the $\delta$ feature is an important factor in finding FNAF. To optimize for the low-dimensional non-differentiable parameter (*i.e.* set of the $(x, y)$ coordinates of $\delta$), we approximate the partial derivatives for each parameter $p$ with the finite central difference:

$$\frac{\partial L}{\partial p} = \frac{L\left(p + \frac{h}{2}\right) - L\left(p - \frac{h}{2}\right)}{h} \tag{6}$$

where $h$ is the step size. Gradient ascent is used to update the location parameter $p$ and maximize Equation (2).

### 3.2. Under-sampling information preservation verification

A benefit of having a synthetic feature generator is that one can quantify the amount of preserved information after k-space under-sampling. To make sure the information of $\delta$ is preserved through under-sampling in the k-space, we make sure the following condition is fulfilled:

$$D(x + \delta, x) < \epsilon \tag{7}$$

where $D$ is a distance function, and $\epsilon$ is a noise error tolerance threshold. We obtain $x + \delta'$ and $x$ through the following:

$$U(y + \delta') = U(y) + U(\delta') = x + \delta \tag{8}$$

as $U$ is linear and closed under addition. MSE is used for $D$.

### 3.3. FNAF-robust training

Our attack formulation allows the reconstruction models to simultaneously undergo standard and adversarial training, while small perturbations-based adversarial training requires models to be trained only on robust features (Madry et al., 2018). This allows one to do FNAF-robust training on a pre-trained model and speed up convergence. To accelerate training, we adopt ideas from Shafahi et al. (2019b): in essence, to do FNAF-robust training, the model utilized a training set which included original and adversarial examples, including the examples that are generated during the search for the worst adversarial case. However, the inner maximization is performed by either random search or finite-difference approximation gradient ascent - described above. Random search reduces our implementation to be a data augmentation approach. Furthermore, strict adversarial training with random search in a worst-of-k fashion like in (Engstrom et al., 2017) might result in improved model robustness, and its implementation in our framework is straightforward.

## 4. Experiments and results

### 4.1. Experimental setup

We conduct our experiments on the FastMRI knee dataset with single-coil setting, including 4x and 8x acceleration factors (Zbontar et al., 2018). We evaluate our methods with two 2-D deep learning based methods, U-Net (Ronneberger et al., 2015) – an popupar baseline, and invertible Recurrent Inference Machines (I-RIM) (Putzky and Welling, 2019) – the winner of the single-coil FastMRI challenge. For U-Net, we follow the training procedures described in Zbontar et al. (2018). For I-RIM we follow the training procedures described in Putzky et al. (2019) and use the official released pre-trained model.

### 4.2. Implementation details

We perform the FNAF attack on the models with a mean-square error (MSE) loss. We constrain the FNAF to comprise 10 connected pixels. The attack mask is placed within the center of a 120-by-120 crop of the image. The constraint ensures that the feature is small and placed in a reasonable location. For random search, 11 randomly shaped FNAF are generated at random locations for each sample in the validation set and the highest adversarial loss is recorded. For finite-difference gradient ascent (FD), we performed the optimizations for the location x and y in 2 iterations. The number of iterations is chosen to have a reasonable computation time and keep the number of forward passes for one sample constant for both methods. The FD step size $h$ is set to 10 and the learning rate to $10^5$.

An attack is rejected when the information-preservation (IP) loss is lower than 0.0001. This is especially important for FNAF-robust training, as we do not want the FNAF-robust model to go to the other extreme and produce hallucination of non-existing features. With regard to FNAF-robust training, the data augmentation approach training procedure described in Section 4.2 is followed. The adversarial loss of Equation (5) is used with *beta* set to 100 to force the model to focus on the small features. To prevent from overfitting in terms of FNAF attack successes, the best model in terms of the standard reconstruction loss on the validation set is selected to be attacked, ignoring the adversarial loss.

### 4.3. Attack evaluation metrics

The average attack loss for the validation set and the attack hit rate are calculated. The average attack loss is defined in Equation (5). An attack is considered to be a hit when the loss is higher than a threshold value $\gamma$. We empirically set $\gamma$ to 0.001, as we observed the FNAF to be mostly lost when the loss is greater than 0.001. The hit rates are conservatively low, as $\gamma$ is set at a high value, so that there might be cases where the FNAF is lost even at loss values below $\gamma$. We speculate that the actual hit rate is likely higher than the value reported in this work.

### 4.4. Attack results

Examples of the FNAF are shown in Figure 1. The result of the attack shown in Table 2 confirms that hypothesis 2 is true in many cases. The attack with FD is weaker than that with random search (RS), which is counter-intuitive. This might be due to various reasons, such as tuning the optimizer hyper-parameters, the number of iterations, etc. Nonetheless, the high success rate of the random search method for both models show that it is fairly easy to find a FNAF in the search space that is heuristically defined. Although I-RIM is more resilient to the attacks than U-Net, the attack rate is still fairly high. This is concerning but also understandable given that deep learning methods are not explicitly optimized for such objective, so these FNAF are at the tail-end of the distribution or even out-of-distribution with respect to the training distribution. Fortunately, we can modify the objective as specified in Section 3.3 to produce a FNAF-robust model which is empirically fairly resilient to the attacks and also has minimal effect in the standard reconstruction quality shown in Table 1.

Table 1: Standard validation set evaluation with SSIM and normalized mean-square error (NMSE)

| 4× | SSIM | NMSE |
|---|---|---|
| U-Net | $0.7213 \pm 0.2621$ | $0.03455 \pm 0.05011$ |
| I-RIM | $0.7501 \pm 0.2546$ | $0.03413 \pm 0.05800$ |
| FNAF-robust U-Net | $0.7197 \pm 0.2613$ | $0.03489 \pm 0.05008$ |

| 8× | SSIM | NMSE |
|---|---|---|
| U-Net | $0.6548 \pm 0.2942$ | $0.04935 \pm 0.04962$ |
| I-RIM | $0.6916 \pm 0.2941$ | $0.04438 \pm 0.06830$ |
| FNAF-robust U-Net | $0.6533 \pm 0.2924$ | $0.04962 \pm 0.05670$ |

### 4.5. Under-sampling information preservation verification

To investigate hypothesis 2, we measure the acceptance rate of the adversarial examples based on the information-preservation loss. Shown in Table 3, a very high acceptance rate is observed across all settings, showing that in most cases the small feature's information is

Table 2: FNAF attack evaluations.

| 4× | RS (Attack Rate %) | FD (Attack Rate %) | RS (MSE) | FD (MSE) |
|---|---|---|---|---|
| U-Net | 84.44 | 72.17 | 0.001530 | 0.001386 |
| I-RIM | 44.49 | 34.60 | 0.001164 | 0.001080 |
| FNAF-robust U-Net | 12.71 | 10.48 | 0.000483 | 0.000466 |

| 8× | RS (Attack Rate %) | FD (Attack Rate %) | RS (MSE) | FD (MSE) |
|---|---|---|---|---|
| U-Net | 86.00 | 74.84 | 0.001592 | 0.001457 |
| I-RIM | 77.39 | 63.88 | 0.001470 | 0.001349 |
| FNAF-robust U-Net | 15.09 | 13.30 | 0.000534 | 0.000467 |

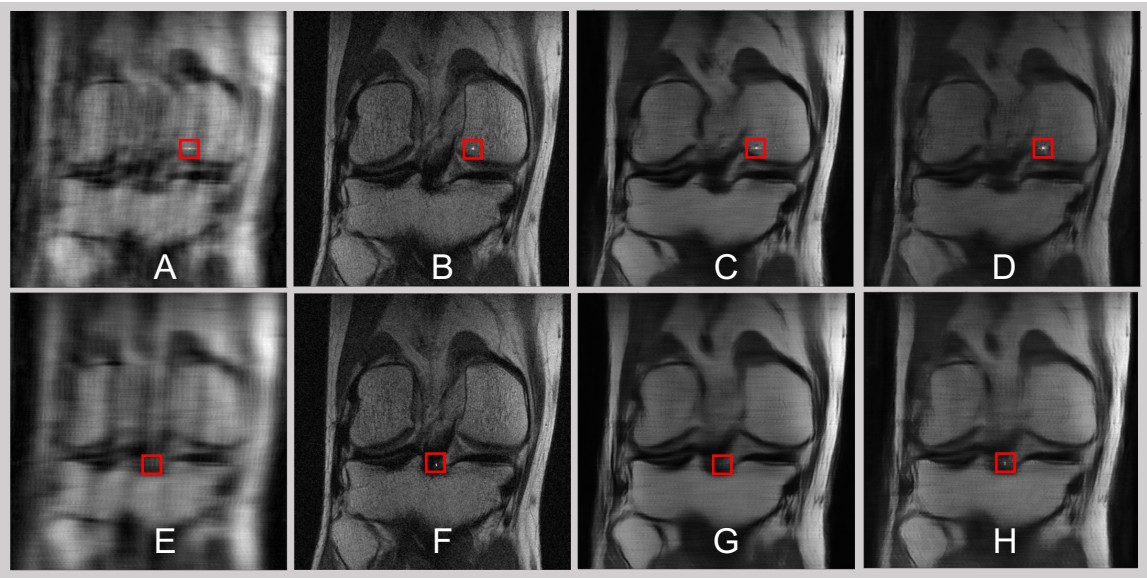

Figure 1: The top row (A-D) shows a "failed" FNAF attack. The bottom row (E-H) shows a "successful" FNAF attack. Column 1 contains the under-sampled zero-filled images. Column 2 contains the fully-sampled ground truth images. Column 3 contains U-Net reconstructed images. Column 4 contains FNAF-robust U-Net reconstructed images. (C-G-D-H) FNAF reconstruction: (C) adversarial loss of 0.000229. (G) adversarial loss of 0.00110. (D) adversarial loss of $9.73 \cdot 10^{-5}$. (H) adversarial loss of 0.000449.

not completely lost through under-sampling, at least for the way we construct the features. We speculate that the same could hold true for real-life abnormalities.

Figure 2 shows a small negative correlation between IP loss and FNAF loss. In fact, we expect that more information would weaken the attack. However, such negative correlation is weak, indicating that there is no strong association. Therefore the preservation of infor-

mation alone cannot predict the FNAF-robustness of the model. So the information loss due to under-sampling is a valid but insufficient explanation for the existence of FNAF.

Table 3: Information preservation

|                       | Random  | U-Net FNAF | I-RIM FNAF | Robust U-Net FNAF |
|-----------------------|---------|------------|------------|-------------------|
| Acceptance Rate (%)   | 99.82   | 99.72      | 99.76      | 99.34             |
| IP Loss (MSE)         | 0.00064 | 0.00050    | 0.00051    | 0.00052           |

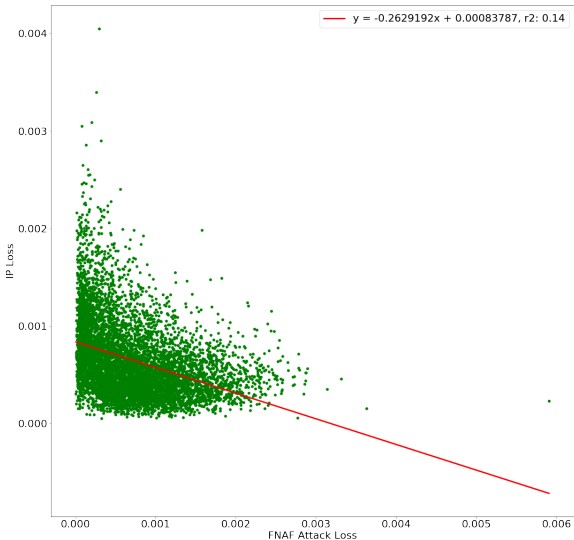

Figure 2: IP loss vs. FNAF loss.

### 4.6. Location distribution of adversarial features

We visualize the location distributions of the worst case FNAF on the image in Figure 3. There seems to be no apparent pattern to the location of the FNAF. However, the location distributions seems to be similar across non-FNAF-robust models. We investigate this in the next section.

### 4.7. Transferability of adversarial features across reconstruction networks

We take FNAF examples from U-Net and apply them to I-RIM, and observe a 89.48% attack rate. The high transferability is similar to what is observed in Goodfellow et al. (2015) and Alcorn et al. (2019). This is indicating that the training data does not capture the distribution of FNAF.

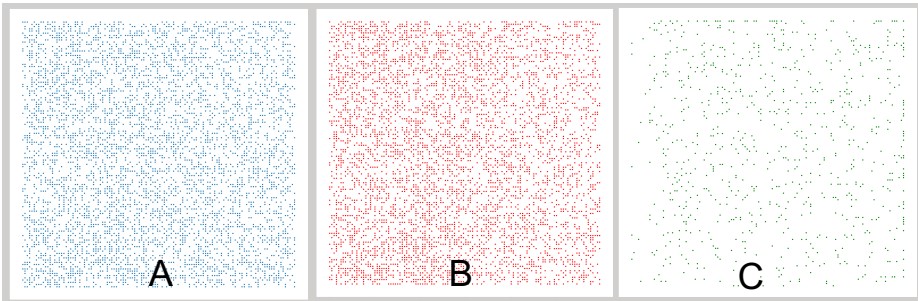

Figure 3: FNAF location distribution within the 120x120 center crop of the image of (A) U-Net, (B) I-RIM, (C) FNAF-robust U-Net

### 4.8. Generalization to real-world abnormalities

A musculoskeletal (MSK) imaging trained M.D. inspects and identifies abnormalities of clinical relevance in 51 volumes from the validation set. The abnormalities include cartilage lesions, meniscal tears, and meniscal degenerations.

The results in Table 4 show that the FNAF-robust U-Net is marginally better out of the small number of abnormalities found. Although further extensive evaluation is needed, this is an encouraging result, considering that there is no guarantee that the synthetic feature would look like real-world abnormalities. The detailed comments of the abnormality findings are included in Appendix A. An example of the results is shown in Figure 4.

We suggest the marginal improvements may be explained by the semantic difference between FNAF and real-world abnormalities, although it certainly requires further investigation. Ideally, we want to construct the space of FNAF to be representative of not only the size but also the semantics of real-world abnormalities. We have two ideas on how one could improve FNAF to be more realistic for future works: 1. Relax the pixel constraint more so that the FNAF space can include real-world abnormalities. 2. Model the abnormality features by introducing domain knowledge. Moreover, it is worth noting that FNAF might not even need to be too realistic for deep learning models to generalize. From our experiments, we observe that by training on our imperfect FNAF, one can force convolution filters to be more sensitive to small features. Overall, we think this is the reason for the observed marginal real-world improvements, and it is indicative of a promising direction to move forward to improve clinical robustness.

## 5. Conclusions

The connection between FNAF to real-world abnormalities is analogous to the connection between lp-bounded adversarial perturbations and real-world natural images. In the natural images sampled by non-adversary, lp-bounded perturbations most likely do not exist. But their existence in the pixel space goes beyond security, as they reveal a fundamental difference between deep learning and human vision (Ilyas et al., 2019). Lp-bounded per-

turbations violate the human prior: humans see perturbed and original images the same. FNAF violate the reconstruction prior: an algorithm should recover (although it may be impossible) all features. We relax this prior to only small features, which often are the most clinically relevant. Therefore, the failure of deep learning reconstruction models to reconstruct FNAF is important even if FNAF might not be representative of the real-world abnormalities. Lp-bounded perturbations inspired works that generate more realistic attacks, and we hope to bring the same interest in the domain of MRI reconstruction. Furthermore, we show the possibility of reconstructing FNAF with adversarial training. In our work, FNAF are constrained to be 10 connected pixels. Arguably, this is smaller than most real-world abnormality features. We believe this indicates that real life abnormalities can be reconstructed from the information preserved through under-sampling.

In this work, we investigate two hypothesis for the false negative problem in deep-learning-based MRI reconstruction. By developing the FNAF adversarial robustness framework, we show that this problem is difficult, but not impossible. Within this framework, there is potential to bring the extensive theoretical and empirical ideas from the adversarial robustness community, especially in the area of provable defenses (Wong and Kolter, 2018; Mirman et al., 2018; Raghunathan et al., 2018; Balunovic and Vechev, 2020) to tackle the problem. We also hope to inspire future work in the direction of defining a better (realistic) search space for the FNAF, towards generalization to real-world abnormalities. We believe that the findings from Appendix A can serve as a validation set for future work.

Table 4: Abnormality Reconstructions

|  | Cartilage Lesion Rate | Meniscus Lesion Rate |
|---|---|---|
| U-Net | 1/8 | 8/9 |
| FNAF-robust U-Net | 3/8 | 9/9 |

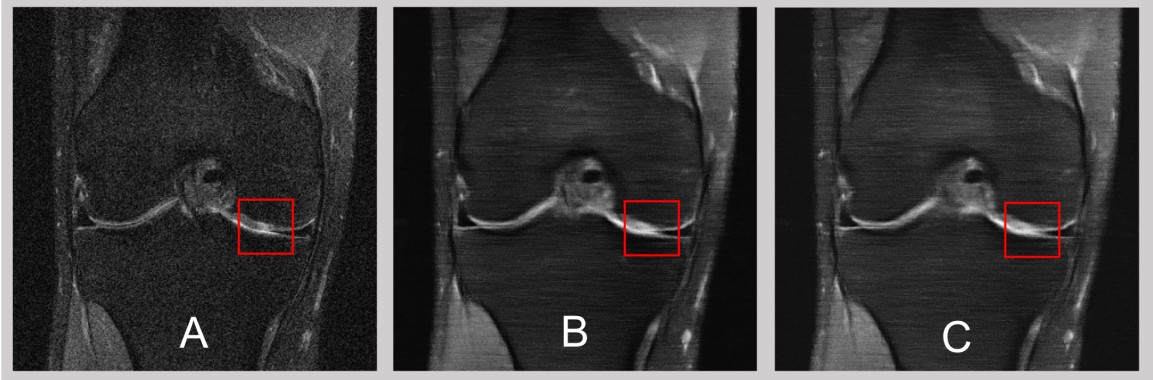

Figure 4: (A) Ground truth: small cartilage lesion in femur. (B) U-Net: Area of cartilage lesion not defined and resembles increased signal intensity. (C) FNAF-robust U-Net: Cartilage lesion preserved but less clear.

## Acknowledgments

We would like to thank Claudia Iriondo for the help with the project and fruitful discussions. We would also like to thank Patrick Putzky and his team for releasing the implementation models for IRIM and pointing us to release page. Ultimately, we would like to thank the reviewers for their constructive feedback and their efforts towards improving our manuscript.

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

# Appendix A. Detailed Comments of Real-World Abnormalities

Table 5: Comments of the MSK radiologist involved in the study. Cases where FNAF-robust U-Net improves compared to U-Net are bolded.

| File | Slice number | Comments on ground truth | Comments on U-Net reconstruction | Comments on FNAF-robust U-Net reconstruction |
|---|---|---|---|---|
| 7 | 27 | Signal change | Original lesion preserved but less clear | **Original lesion preserved** |
| 26 | 16 | Cartilage lesion | Cartilage lesion in original now looks like signal change | Cartilage lesion in original now looks like signal change |
| 52 | | Metal artifacts in tibia | No change in metal artifacts | Metal artifacts preserved |
| 71 | 23 | Cartilage lesion in tibia | Original cartilage lesion not seen | **Original cartilage lesion preserved but less clear** |
| 73 | 23 | Intrasubstance degeneration | Intrasubstance degeneration preserved | Intrasubstance degeneration preserved |
| 107 | 16 | Cartilage lesion in femur | Original cartilage lesion not seen | Original cartilage lesion not seen |
| 114 | 26 | Vertical tear in meniscus | Original tear preserved but less clear | Original tear preserved but less clear |
| 178 | 14-21 | Menisectomy | Menisectomy preserved | Menistectomy preserved |
| 196 | 26 | Horizontal meniscal tear | Meniscal tear preserved | Meniscal tear preserved |
| 201 | 25 | Signal change in femoral cartilage | Cartilage lesion not preserved | **Cartilage lesion preserved but less clear** |
| 267 | 24 | Meniscal tear | Original tear preserved but less clear | Original tear preserved but less clear |
| 280 | 22 | Cartilage lesion in tibia | Original lesion not preserved | Original lesion not preserved |
| 314 | 14-20 | Meniscal degeneration/menisectomy | Meniscal degeneration preserved | Meniscal degeneration preserved |
| 325 | 24 | Signal change in cartilage | Original cartilage lesion not preserved | **Signal change in cartilage partially preserved** |
| 356 | 21 | Cartilage lesion | Original lesion preserved but not clear | **Original cartilage lesion preserved** |
| 464 | 26 | Intrasubstance degeneration | Intrasubstance degeneration preserved but not clear | **Intrasubstance degeneration preserved** |
| 480 | 21 | Cartilage lesion | Cartilage lesion not preserved | Cartilage lesion not preserved |
| 528 | 28 | Intrasubstance degeneration | Intrasubstance degeneration not preserved | **Intrasubstance degeneration preserved** |

