# OpenReview forum: "Addressing The False Negative Problem of MRI Reconstruction Networks by Adversarial Attacks and Robust Training"
_MIDL.io/2020/Conference — MIDL 2020_

### Official Review · AnonReviewer1 · 2020-03-12
**Investigation of two hypothesis for the false negative problem in deep-learning-based MRI reconstruction.**

**Rating:** 2
**Confidence:** 2

**Summary:**

The paper investigates two hypothesis for the false negative problem in deep learning based MRI reconstruction.  The small and rare features are the most relevant in clinical diagnostic settings. The authors develop the FNAF adversarial robustness framework. Using the proposed frame work the worst case false negatives by the adversarial attacks are found and used to improve the reconstruction task.


**Strengths:**


The paper proposes a new idea to improve the robustness of the networks for reconstructing the MR images. The relevant clinical application is also interesting.  The results are validated using several metrics.

**Weaknesses:**

The contributions of the paper are not clear and It's not easy to follow the text. The paper proposes an idea for robustness of the network but according to the results the improvement is marginal. It is not clear how this adversarial perturbations are generalisable to the real world data.

**Justification Of Rating:**

The contributions of the paper are not clear and It's not easy to follow the text. The paper proposes an idea for robustness of the network but according to the results the improvement is marginal. It is not clear how this adversarial pertubations are generalizable to the real world data.

**Paper Type:**

methodological development

**Special Issue:**

no

---

> ### Author Response · Authors · 2020-03-25
> **Clarification of contribution and connections between FNAF and real-world abnormalities**
>
> We would first like to thank the reviewer’s positive comments regarding the novelty, clinical relevance, and the good results under synthetic setup.
>
> Q1 : The contributions of the paper are not clear.
>
> A1: A summary of the contributions:
> We quantitatively showed that the best deep learning reconstruction models trained to only maximize image quality can fail to reconstruct small and infrequent structures. The transferability of false-negative adversarial features (FNAF) between models suggests that FNAF are similar to the l-p norm bounded adversarial perturbation in the computer vision community. According to theoretical works (Ilyas et al. (2019), Shafahi et al. (2019)), adversarial examples are hard to avoid without having a defense mechanism in place. This means that without specific priors, it might be hard for a deep learning reconstruction model to reconstruct small and infrequent abnormalities, to be deployed in the actual clinical setting. We believe having a transferable, scalable (no labeling needed) and quantitative method for testing the clinical robustness of reconstruction models is as important as solving the false negative problem, as the community can use this as a benchmark for clinical relevance which goes beyond mere image quality in future works.
> We quantitatively showed that it is possible to reconstruct small structures if the right set of priors is available during training, particularly if adversarial training is employed. This corresponds to the second hypothesis in the paper. This is an important finding as we address the over-pessimism that no reconstruction models can reconstruct these features, as necessary information is lost when under-sampling.
>
> Q2: It is not clear how this adversarial perturbations are generalisable to the real world data.
>
> A2: We agree it is important to make a connection between FNAF and real-world abnormalities clear in the paper. We relate the connection back to our contributions:
> For what concerns contribution 1, we think the connection between FNAF to real-world abnormalities is analogous to the connection between lp-bounded adversarial perturbations and real-world natural images. In the natural images sampled by non-adversary, lp-bounded perturbations most likely do not exist. But their existence in the pixel space goes beyond security, as they reveal a fundamental difference between deep learning and human vision (Ilyas et al. (2019)). Lp-bounded perturbations violate the human prior: humans see perturbed and original images the same. FNAF violate the reconstruction prior: an algorithm should recover (although it may be impossible) all features. We relax this prior to only small features, which often are the most clinically relevant. Therefore, the failure of deep learning reconstruction models to reconstruct FNAF is important even if FNAF might not be representative of the real-world abnormalities. lp-bounded perturbations inspired works that generate more realistic attacks, and we hope to bring the same interest in the domain of MRI reconstruction.
> With regard to contribution 2, we showed the possibility of reconstructing FNAF with adversarial training. In our work, FNAF are constrained to be 10 connected pixels. Arguably, this is smaller than most real-world abnormality features. We believe this indicates that real life abnormalities are possible to can be reconstructed from the information preserved through under-sampling.
>
> Q3: The paper proposes an idea for robustness of the network but the improvement is marginal.
>
> A3: We suggest the marginal improvements may be explained by the semantic difference between FNAF and real-world abnormalities, although it certainly requires further investigation. Ideally, we want to construct the space of FNAF to be representative of not only the size but also the semantics of real-world abnormalities.
> We have two ideas on how one could improve FNAF to be more realistic for future works: 1. Relax the pixel constraint more so that the FNAF space can include real-world abnormalities. 2. Model the abnormality features by introducing domain knowledge. Moreover, it is worth noting that FNAF might not even need to be too realistic for deep learning models to generalize. From our experiments, we observed that by training on our imperfect FNAF, one can force convolution filters to be more sensitive to small features. Overall, we think this is the reason for the observed marginal real-world improvements, and it is indicative of a promising direction to move forward to improve clinical robustness.
>
> Q4: It's not easy to follow the text
> A4: We agree that portions of the manuscript require rewriting, we will refine them in the revised version.
>
> Citations:
> [1] Andrew Ilyas et al. Adversarial examples are not bugs, they are features. Advances in Neural Information Processing Systems, 2019.
> [2] Ali Shafahi, et al. Are adversarial examples inevitable? In International Conference on Learning Representations, 2019a.

---

### Official Review · AnonReviewer2 · 2020-03-13
**In a conference like MIDL, more importance should be given to formulation**

**Rating:** 2
**Confidence:** 3

**Summary:**

This paper present an application specific approach to obtain more robust reconstruction deep learning models for MRI. The goal of the reconstruction model is to reconstruct full MRIs from sub-sampled MRIs. To obtain a more robust model, artificial features are added to the images. These adversarial features are constructed such that they maximize the reconstruction model error.

**Strengths:**

- The paper presents an interesting approach to improve reconstruction model’s robustness by leveraging domain specific knowledge.
- The results, especially Table 5, look promising as more visual features were preserved using the robust model.


**Weaknesses:**

- Many details are missing in the paper, making it difficult to understand what was done precisely. Given that some sections are unnecessary (2.3 and 2.4), there was room to put these details.
- The training protocol with random search is data-augmentation and should be presented as such. This data augmentation is simply domain specific.
- The quantitative results seem incomplete and/or wrong: it is mentioned that the experiments are conducted on 4x and 8x accelerations but the Table 1 seems to report only 8x and the reported I-RIM results are incorrect if it is 8x acceleration (from the I-RIM paper).
- Too much importance is given to some parts like Figure 2 given that the paper is already longer than 8 pages and misses other important details.

**Detailed Comments:**

- Section 2.3 and 2.4 are unnecessary.
- The notations should be formally defined at the beginning of section 3.
- Equation 1 (and preceding sentences) corresponds to only one type of adversarial attack. Saying that this formalizes “the adversarial attack” is a bit misleading to a reader not familiar with adversarial attacks.
- I feel like the under sampling function and the k-space mask function need more explanations.
- If one wants to make sure than enough information is preserved, then one needs $D(x+\delta’,x)<\epsilon$ and not the opposite as in equation 5.
- $\delta$ is typically an optimized perturbation added to $x$ in adversarial literature, but then it is called a feature in Section 3.3 and is fixed. This should be given another name given that the paper talks about adversarial examples.
- “The adversarial loss is weighted by 100”: Does this mean there is a second loss? What is it?
- Information-preservation loss is not defined anywhere except in a table between brackets.


**Justification Of Rating:**

Overall, this paper presents an interesting research direction and gives some evidence that the proposed strategy may help obtaining better reconstruction models for MRI. However, the paper is missing many details of formulation and implementation which are necessary in a conference like MIDL.

**Paper Type:**

methodological development

**Questions To Address In The Rebuttal:**

- “The loss was adapted from Kervadec et al. (2018) and Caliv´a et al. (2019): we only keep the foreground and a small portion of the background with respect to the border of the synthetic feature.” could you elaborate on that? This loss (reference to Caliva et al. is unnecessary here) does not discard a part of the input. This should be detailed.
- “We further adopt ideas from Shafahi et al. (2019) to accelerate training.” Which ideas?
- It is unclear from the paper if regular adversarial training (with PGD) is performed. I would like clarifications on that.


**Special Issue:**

no

---

> ### Author Response · Authors · 2020-03-28
> **Response 1**
>
> We thank the reviewer for acknowledging our appropriate use of domain knowledge. We would also like to thank them for their constructive comments they provided to improve the paper’s clarity.
>
> Q: $\delta$ is typically an optimized perturbation added to  in adversarial literature, but then it is called a feature in Section 3.3 and is fixed. This should be given another name given that the paper talks about adversarial examples.
>
> A: We consider $\delta$ to be adversarial in terms of degrading the performance of the task; this is consistent with the adversarial literature. In our formulation, $\delta$ can be called a perturbation as it is added to the original input synthetically. However, we also call $\delta$ a feature, or false-negative adversarial features (FNAF), since they are visible. FNAF are not fixed; they are optimized and added to the input, so that the loss is maximized.
> We understand that the use of $\delta$ for FNAF might cause confusion, therefore we will utilize an alternative notation in the revised manuscript.
>
> Q:
> - “The loss was adapted from Kervadec et al. (2018) and Caliv´a et al. (2019): we only keep the foreground and a small portion of the background with respect to the border of the synthetic feature.” could you elaborate on that? This loss (reference to Caliva et al. is unnecessary here) does not discard a part of the input. This should be detailed.
> - “The adversarial loss is weighted by 100”: Does this mean there is a second loss? What is it?
>
> A: We agree, the description of the loss may require further details to improve its comprehension. Our loss $L$ is defined as
> $$L(a, b) = \alpha \text{MSE}(a, b) + \beta \text{MSE}(mask(a), mask(b)) $$
> where a and b are the original MRI and the reconstructed MRI respectively. $mask$ is an indicator function which masks over the FNAF in the ground truth and the reconstructed image. Weights $\alpha$ and $\beta$ are hyperparameters set to 1 and 100 during adversarial training. This allows us to better preserve both image quality and robustness of FNAF. Conversely, during attack evaluation, $\alpha$ and $\beta$ are set 0 and 1, to better evaluate the reconstruction of FNAF.
> The mask also accounts for the feature boundary, the reason for which we initially cited Kervadec et al. (2018) and Caliva et al. (2019). However, we agree that since the core idea between these and our work are different, we will remove these citations in the revised manuscript. The motivation for the mask accounting for boundaries is: if we only measured the loss of the FNAF’s foreground, this might not capture some failure cases where the FNAF had blended in with the background. Therefore, in addition to the foreground of FNAF, we also measure the loss in pixels that are $r$ pixels away from the boundary of FNAF. We set $r$ to be small, 5.
> The “adversarial loss” in the paper only referred to the $\beta$ weighted part of the loss which we understand might have caused confusion. We will address this in the amended manuscript .
>
>
> Q: - The training protocol with random search is data-augmentation and should be presented as such. This data augmentation is simply domain specific.
> - “We further adopt ideas from Shafahi et al. (2019) to accelerate training.” Which ideas?
> - It is unclear from the paper if regular adversarial training (with PGD) is performed. I would like clarifications on that.
>
> A: The FNAF adversarial training can be described as follows:
> - for each training example, find the worst cases of FNAF (= cases that maximize the loss), and train with these worst-case examples. This is similar to pgd adversarial training. Differing from pgd, our inner maximization is performed by either random search or finite-difference approximation gradient ascent. From Shafahi et al. (2019), we take the idea that we can also train with the examples that were used to find the worst-case, as opposed to using only the worst-case, for faster training. In the case of random search, the implementation is reduced to data augmentation. However, strict adversarial training with random search in a worst-of-k fashion like Engstrom et al. (2019) might result in better robustness performance, and its implementation in our framework is straightforward.

---

> ### Author Response · Authors · 2020-03-28
> **Response 2**
>
> Q: The quantitative results seem incomplete and/or wrong: it is mentioned that the experiments are conducted on 4x and 8x accelerations but the Table 1 seems to report only 8x and the reported I-RIM results are incorrect if it is 8x acceleration (from the I-RIM paper).
>
> A: We could not replicate the results of I-RIM because their implementation for FastMRI had not been released at the time of submission and the implementation included additional details to those in their manuscript or talk. Recently, their official implementation and pretrained model were released, hence we applied their model to our problem. While in the manuscripts, all models were trained and evaluated using a mix of 4x and 8x acceleration factors, here we report the performance on 4x and 8x separately, of the models which were trained with a mixed of 4x and 8x.
>
> Standard validation set evaluation (SSIM)
> AF |U-Net       |I-RIM       |FNAF-robust U-Net
> 4x  |0.7213±0.2621|0.7501±0.2546|0.7197±0.2613
> 8x  |0.6548±0.2942|0.6916±0.2941|0.6533±0.2924
>
> Standard validation set evaluation (NMSE)
> AF |U-Net                   |I-RIM                   |FNAF-robust U-Net
> 4x  |0.03455±0.05011|0.03413±0.05800|0.03489±0.05008
> 8x  |0.04935±0.05697|0.04438±0.06830|0.04962±0.05670
>
> FNAF attack evaluations with RS (Attack rate %)
> AF |U-Net  |I-RIM    |FNAF-robust U-Net
> 4x  |84.44   |44.49   |12.71
> 8x  |86.00   |77.39   |15.09
>
> The I-RIM model that has better standard reconstruction performance is more resistant to FNAF attacks especially under 4x setting, but not better than the FNAF robust U-Net. The results are still consistent with our original conclusions.
>
> Q: Many details are missing in the paper, making it difficult to understand what was done precisely. Given that some sections are unnecessary (2.3 and 2.4), there was room to put these details.
>
> A: We will add the details addressed during the rebuttal process and condense/cut section 2.3 and 2.4.
>
> Q: Too much importance is given to some parts like Figure 2 given that the paper is already longer than 8 pages and misses other important details.
>
> A: We think Figure 2 is helpful to understand that FNAF is not entirely correlated with the information preserved through undersampling. We agree the section needs to be summarized to allow for more room for the details related to the rebuttal.
>
> Q:The notations should be formally defined at the beginning of section 3.
>
> A: We will add formal definitions of the notations at the beginning of section 3 in the revised version of the paper.
>
> Q: Equation 1 (and preceding sentences) corresponds to only one type of adversarial attack. Saying that this formalizes “the adversarial attack” is a bit misleading to a reader not familiar with adversarial attacks.
>
> A: We think that Equation 1 formulizes all known adversal attacks in the supervised learning setting including ours, as the space of $\delta$ can be within $l-p$ norm, the semantic space, the physical etc. In our case, $\delta$ is in the FNAF space, and one can define $y$ in Equation 1 to be $y+\delta$ even though $\delta$ is not fixed which means $y+\delta$ is not fixed. We will add this clarification to the revised paper..
>
> Q: I feel like the under sampling function and the k-space mask function need more explanations.
>
> A: The under-sampling function and the k-space mask function are the same as the implementations from Zbontar et al. (2018)
>
> Q: If one wants to make sure than enough information is preserved, then one needs $D(x+\delta^{\prime}, x) < \epsilon$ and not the opposite as in equation 5.
>
> A: We apologise for the typo in the paper. As correctly pointed out, the equation should be $D(x+\delta^{\prime}, x) < \epsilon$.
>
> Q: Information-preservation loss is not defined anywhere except in a table between brackets.
>
> A: We will define information-preservation loss as MSE in the main text for the revision.
>
> We appreciate all the constructive comments and will include the details addressed here in the revision.
>
> Citations:
> [1] Logan Engstrom et al. Exploring the landscape of spatial robustness. In International Conference on Learning Representations, 2019.
> [2] Jure Zbontar, et al. fastmri: An open dataset and benchmarks for accelerated mri. arXiv preprint arXiv:1811.08839, 2018.

---

### Official Review · AnonReviewer3 · 2020-03-14
**How to prevent "false negatives" in deep learning based reconstruction.**

**Rating:** 4
**Confidence:** 5
**Recommendation:** Oral

**Summary:**

This is an interesting paper addressing the "false negatives" (i.e. pathologies which are not properly reconstructed) in deep learning-based MRI reconstruction. During the MedNeurIPS conference, the organizers of the FastMRI challenge have shown that even the best algorithms have considerable disadvantages: they are able to remove pathologies such as meniscal tears and subchrondral osteophytes in the reconstructed image. This is clearly a disadvantage and limits the acceptance of these techniques, and the authors attempt to solve this by using adversarial techniques to improve reconstruction.

**Strengths:**

* The authors hypothesise that there can be two reasons pathologies are not properly reconstructed: as they are small they might be in the higher frequencies and therefore more likely to be filtered by the sampling scheme (which was my assumption) or they are not properly reconstructed. The authors give evidence towards the latter, which is kind of surprising!
* The authors show that adversarial training by inserting "hotspots" of a few voxels into the image helps in reconstructing the above mentioned artefacts
* Radiologist involved in the study
* Tested both the iRIM and the U-net reconstruction
* Results are not that affected much in terms of SSIM and PSNR.

**Weaknesses:**

* The paper is not that clearly written. To begin with, the community surely knows what an adversarial example is as everyone will know the examples of having stickers on road signs giving completely different predictions, but it is harder to understand in this context. The link with adversarial examples is not that obvious. You might want to consider rewriting it a bit and mention that these ideas are inspired by adversarial examples. For instance, what is the relevance of 2.3?
* The loss is L2 while (iirc) all the winning solutions used a SSIM loss or L1 + SSIM. This is not the same as the solution of the FastMRI challenge as with the iRIM the models all perform better when using L1/SSIM. Perhaps this method works better with L2?


**Justification Of Rating:**

The paper is excellent in content, and even though it can definitely be improved by rewriting it significantly, this does not affect my score as this addresses a very important question and helps the field of machine learning reconstruction in MRI make a step forward.

**Paper Type:**

methodological development

**Questions To Address In The Rebuttal:**

This is an excellent paper. However, it would be very beneficial if it is a bit rewritten. The linking between adversarial attacks is not that clear and the flow is a bit missing in the paper. I definitely enjoyed reading it and it is a proper step forward.

**Special Issue:**

yes

---

> ### Author Response · Authors · 2020-03-28
> **Response**
>
> We would like to thank the positive reviews and the constructive comments to improve the paper.
>
> Q: The paper is not that clearly written. To begin with, the community surely knows what an adversarial example is as everyone will know the examples of having stickers on road signs giving completely different predictions, but it is harder to understand in this context. The link with adversarial examples is not that obvious. You might want to consider rewriting it a bit and mention that these ideas are inspired by adversarial examples. For instance, what is the relevance of 2.3?
>
> A: Thank you for the suggestion to clarify for the community. We consider an adversarial example to be: an example that after a certain perturbation is added, maximizes a certain loss function. This is formulated by Equation 1. This definition generalizes to all adversal attacks in the Section 2 and FNAF attack. For an adversarial attack to be meaningful, one needs to carefully craft the space of perturbation and the loss function that makes sense in the context of the task, which often involves human priors. For the example of stickers on road signs, the perturbation space is chosen as this might occur in the real world, and the loss function is chosen to be the original loss function with original label, as the labels would not change for a human. For FNAF, the perturbation is an unsampled visible small feature, as it could resemble an abnormality, and the loss function is the reconstruction loss of the visible small feature, as we want the reconstruction algorithm to reconstruct the visible small feature.
>
> Q: The loss is L2 while (iirc) all the winning solutions used a SSIM loss or L1 + SSIM. This is not the same as the solution of the FastMRI challenge as with the iRIM the models all perform better when using L1/SSIM. Perhaps this method works better with L2?
>
> A: Thank you for the suggestion to investigate different loss functions. The loss function for adversarial attack and adversarial training can be any reconstruction loss function in our framework. We did not do a comparison between the loss functions as this is not central to the message of this paper. We choose L2 loss abitrially. It is possible the SSIM is a better loss for generalizing to real-world abnormalities. We wish to investigate this in future works.

---

### Official Review · AnonReviewer4 · 2020-03-18
**Interesting approach to use adversarial learning to ensure small feature survival in accelerated/undersampled MRI reconstruction**

**Rating:** 4
**Confidence:** 4
**Recommendation:** Oral

**Summary:**

key ideas: The basic idea of the paper is to use adversarial learning to ensure small feature survival in accelerated/undersampled MRI reconstruction. Essentially the clinical issue is that when looking at the best and worst performing reconstruction models with deep learning in terms of qualitative measures (SSIM) and radiologists' image quality assessment, these models fail in reconstructing some relatively small abnormalities. These are then defined as false negatives. The paper tries to mitigate the miss of these by employing adversarial learning.

experiments: The experiments are performed on the well-established OpenSource fastMRI dataset that consists of knee MRI images with a single-coil setting, including 4x and 8x acceleration factors. The authors embed synthetic '`false-negative adversarial feature' (FNAF), a perceptible small feature that is present in the ground truth MRI but has disappeared upon MRI reconstruction via a learning model, in their data and see if models (U-net and I-RIM) improve in the FNAF retrieval under attack. While the approach is very interesting and also seems to work on the synthetic data in terms of achieving the same quantitative measures (SSIM and PSNR) with a lower attack rate, on real data aka compared to a radiologist the results in Table 4 show that the FNAF-robust U-Net is only marginally better out of the small number of abnormalities found.

significance: Dealing with accelerated MRI is a very important clinical problem and especially the issue of missing small features is of even bigger clinical importance. Hence I find this paper makes an important contribution to the field.

**Strengths:**

-The paper is well written and the problem as well as its clinical importance clearly stated
- The method section is extensive and hence also clear
-The synthetic evaluation is done carefully and with the real world scenario in mind


**Weaknesses:**

-While the approach is very interesting and also seems to work on the synthetic data in terms of achieving the same quantitative measures (SSIM and PSNR) with a lower attack rate, on real data aka compared to a radiologist the results in Table 4 show that the FNAF-robust U-Net is only marginally better out of the small number of abnormalities found.
- In general experiments and results fall a bit short, there's no explicit discussion
- The generalization to real-world abnormalities is really what is missing. The authors do not discuss this much, but point to it as future work.

**Detailed Comments:**

se above

**Justification Of Rating:**

As stated above dealing with accelerated MRI is a very important clinical problem and especially the issue of missing small features is of even bigger clinical importance. Hence I find this paper makes an important contribution to the field. Even though the evaluation on the real data set aka compared to a radiologist has until now only marginal improvements, this is something worthwhile to look at.

**Paper Type:**

both

**Questions To Address In The Rebuttal:**

- In general, your introduction is nice and extensive and so is the methods section, but I find the Experiments and Results rather short. It might be worth to shorten some of the start and then conversely extend the discussion part of the paper. By the following points:
1) what are the drawbacks of the features (FNAFs) you now simulated?
2) address the generalization to real-world abnormalities further. You only mention in 4.8 some of the results, that section should be expanded and you should further discuss why the models fail.

- Also, the size of the FNAFs should also be correlated to the undersampling factor, since if the image features are very small with a high undersampling factor more of the high frequency information will be missing. You mention that you use both 4x and 8x acceleration but don't state anywhere if the models behave differently under different accelerations.

**Special Issue:**

yes

---

> ### Author Response · Authors · 2020-03-28
> **Response**
>
> We would like to thank the positive reviews and the constructive comments to improve the paper.
>
> Q: What are the drawbacks of the features (FNAFs) you now simulated? discuss why the models fail.
>
> A: Similar to our response to reviewer 1, we suggest the marginal improvements may be explained by the semantic difference between FNAF and real-world abnormalities, although it certainly requires further investigation. Ideally, we want to construct the space of FNAF to be representative of not only the size but also the semantics of real-world abnormalities. Also, our FNAF is constrained to be only 10 pixels and has high intensity for the simplicity of the experiments, which might not be a space that includes all abnormalities. This is the drawbacks of our constrained FNAF.
>
>
> Q: Address the generalization to real-world abnormalities further.
>
> A: Similar to our response to reviewer 1, we have two ideas on how one could improve FNAF for future works: 1. Relax the constraints more so that the FNAF space can include real-world abnormalities. 2. Model the abnormality features by introducing more domain knowledge.
> We lean towards the former as it is more generalized and fits more nicely into the robust training direction: if the perturbation space includes all abnormalities, then we can achieve real world robustness guarantee through adversarial training, or provable defenses.
>
>
> Q: Also, the size of the FNAFs should also be correlated to the undersampling factor, since if the image features are very small with a high undersampling factor more of the high frequency information will be missing. You mention that you use both 4x and 8x acceleration but don't state anywhere if the models behave differently under different accelerations.
>
> A: Thank you for the suggestion to investigate the correlation between the undersampling factor and FNAF. By the size of the FNAFs, do you mean the occurrence of FNAFs? If so, your hypothesis is in line with the results we present below. In the manuscripts, all models were trained and evaluated using a mix of 4x and 8x acceleration factors. Here, we present the evaluations of the 4x-8x-mixed-trained models with 4x and 8x separately. I-RIM results is changed from our implementation to the official pre-trained models from Putzky et al (2019) released after the submission of this paper:
>
> Standard validation set evaluation (SSIM)
> AF |U-Net       |I-RIM       |FNAF-robust U-Net
> 4x  |0.7213±0.2621|0.7501±0.2546|0.7197±0.2613
> 8x  |0.6548±0.2942|0.6916±0.2941|0.6533±0.2924
>
> Standard validation set evaluation (NMSE)
> AF |U-Net                   |I-RIM                   |FNAF-robust U-Net
> 4x  |0.03455±0.05011|0.03413±0.05800|0.03489±0.05008
> 8x  |0.04935±0.05697|0.04438±0.06830|0.04962±0.05670
>
> FNAF attack evaluations with RS (Attack rate %)
> AF |U-Net  |I-RIM    |FNAF-robust U-Net
> 4x  |84.44   |44.49   |12.71
> 8x  |86.00   |77.39   |15.09
>
> The I-RIM model that has better standard reconstruction performance is more resistant to FNAF attacks especially under 4x setting, but not better than the FNAF robust U-Net. The results are still consistent with our original conclusions.

---

### Meta-Review · Area_Chair1 · 2020-04-06
**MetaReview of Paper28 by AreaChair1**

**Rating:** 4
**Recommendation For Accepted Papers:** Best Paper Award, Oral

**Metareview:**

This paper proposes an adversarial attack strategy to overcome the problem with accelerated MRI models missing small, rare (but often important!) features. The paper is interesting, tackles an important problem, and reaches interesting conclusions on the source of the problem.

The reviewers also point out that the paper is difficult to read at times; the authors are strongly encouraged to take the feedback from these reviews into account to make an even better camera ready paper.

**Paper Type:**

both

**Special Issue:**

yes

---

### Decision · Program_Chairs · 2020-04-11

Accept